# Reproducibility of the German and Slovakian Versions of the Dietary Habits and Nutrition Beliefs Questionnaire (KomPAN)

**DOI:** 10.3390/nu14224893

**Published:** 2022-11-19

**Authors:** Elżbieta Cieśla, Edyta Suliga, Helena Kadučáková, Sven Michel, Marcela Ižová, Viera Simočková, Titus Martin, Alexander Braun, Dorota Kozieł, Stanisław Głuszek

**Affiliations:** 1Institute of Health Sciences, Medical College, Jan Kochanowski University, 25-369 Kielce, Poland; 2Department of Nursing, Faculty of Health, Catholic University in Ružomberok, 03401 Ružomberok, Slovakia; 3Institute for Health, Faculty of Social Work, Health and Music, Brandenburg University of Technology Cottbus-Senftenberg, 03048 Cottbus, Germany; 4Institute of Medical Sciences, Medical College, Jan Kochanowski University, 25-369 Kielce, Poland

**Keywords:** food frequency questionnaire, dietary habits, reproducibility

## Abstract

Purpose. The aim of this study was to assess the reproducibility and reliability of the KomPAN questionnaire among two groups of university students from Germany and Slovakia. Methods. A total of 422 individuals (mean age 21.4 years, SD 4.0), including 197 from Slovakia (men 26.2%) and 225 from Germany (men 22.3%), were tested using the self-administered (SA-Q) version of the KomPAN questionnaire and then retested two weeks later. A cross-classification analysis, kappa coefficients, Cronbach’s ɑ coefficients, and a test-retest result comparison were conducted separately for each group of students to assess the reproducibility and reliability of the questionnaire. Results. The cross-classification values were higher than 46.2% among the German students and higher than 55.8% among the Slovakian students. The kappa coefficients ranged from 0.21 to 0.90 in the German students and from 0.38 to 0.94 in the Slovakian students. Cronbach’s ɑ ranged from 0.58 to 0.78. Conclusion. The questionnaire displayed a moderate to very good reproducibility, which was slightly higher in the Slovakian group than in the German group. Therefore, the questionnaire can be recommended for further analysis and comparison of the dietary habits among Germans and Slovakians on a larger scale.

## 1. Introduction

Dietary habits are behaviours related to choosing and consuming food. Due to economic and cultural differences in the usual foods and traditional meals between countries [1,2,3], it is important to develop universal tools that are applicable in international studies.

A highly popular method in nutritional epidemiology studies is to assess the frequency of consumption of different food products [4,5,6]. This method is relatively easy to implement, as well as being cheap and useful in interpopulation comparative studies. Various questionnaires have been developed to this effect. The products included in these questionnaires are usually limited to the most commonly consumed foods and the sources of basic nutrients. The obtained responses allow researchers to assess long-term dietary habits fairly and accurately and to categorise the study participants into groups with a low, medium, or high intake of particular products, in order to investigate the causal relationships between the consumption of food and the development of chronic diseases. Some questionnaires can also be used to assess the intake of nutrients or food groups [7,8]. Newly-developed questionnaires must be validated to ensure their high quality [9,10,11,12]. Already established and validated questionnaires must also undergo validation when they are planned to be used in different conditions or among a different population. The reproducibility of a given method is evaluated using the test-retest procedure, i.e., by comparing the results obtained with that method to those obtained when reapplying it later among the same group. The satisfactory reproducibility of a questionnaire is especially important in studies concerning the relationship between nutrition and the development of diet-dependent diseases that are aimed at indicating risk groups [6,13].

In order to standardise the analyses of nutrition conducted by many different researchers and establish a domestic standard, the Dietary Habits and Nutrition Beliefs Questionnaire (KomPAN) for individuals aged 15–65 years was developed in Poland [14]. A validation of the questionnaire conducted among the Polish population indicated a moderate to very good reproducibility [15]. Consequently, the questionnaire was accepted as appropriate for the assessment of dietary habits, lifestyle, and nutritional awareness among teenagers and adults, and among both healthy individuals and those with chronic diseases [15,16]. The aim of this study was to validate the German and Slovakian language versions of the KomPAN questionnaire by evaluating its reproducibility.

## 2. Material and Methods

Between 2020 and 2022, a study was conducted aiming at assessing the eating habits and physical activity among students at three universities: the Jan Kochanowski University in Kielce, Poland; the University of Ružomberok, Slovakia; and the Brandenburg University of Technology Cottbus-Senftenberg, Germany. For the universities in Slovakia and Germany, a test-retest procedure was applied for the assessment of the students’ nutritional status in order to validate the self-administered (SA-Q) version of the KomPAN. The retest took place two weeks after the test among the same groups of students. In Slovakia, the study was conducted between November 2020 and February 2021 (*N* = 231). In Germany, due to the SARS-CoV-2 pandemic, the study was conducted into two stages: November 2020 to January 2021 (*N* = 119), and November 2021 to January 2022 (*N* = 121). Three inclusion criteria were used in this study: university students of medical courses except medicine and dietetics, declared willingness to participate in the test and retest, and no disorders requiring a special diet. A total of *N* = 49 respondents were removed from the dataset due to incomplete data, including *N* = 15 respondents from Germany and *N* = 34 respondents from Slovakia. In total, 422 complete questionnaires filled in by students studying nursing, public health, and physiotherapy qualified for the analysis, including 197 students from Slovakia and 225 from Germany (Figure 1).

This study was approved by the Bioethics Committee of the Jan Kochanowski University in Kielce (Approval No. 10/2015 of 8 April 2015) and was performed in accordance with the ethical standards laid out in the 1964 Declaration of Helsinki and its later amendments. 

The Dietary Habits and Nutrition Beliefs Questionnaire for individuals 15–65 years old (KomPAN), developed by the Behavioural Nutrition Team, Committee of Human Nutrition, Polish Academy of Sciences, consists of four parts with thematically grouped questions [13]. The first part allows for a comprehensive characterisation of dietary habits (e.g., the number of meals consumed per day or snacking between meals). It contains 11 questions, 3 of which are multiple (questions: 10—snacking between meals, 12—type of meat consumed most frequently, and 17—type of water consumed). These questions are encoded separately as dichotomous questions with two possible answers: ‘yes’ or ‘no’. The other questions in this part concern the number and regularity of meals consumed, snacking, drinking sweetened beverages, adding salt to meals, heat treatment of meat, and fats used to prepare meals. The number of categories for these questions ranges from 3 to 6. The second part of the questionnaire assesses the respondent’s food frequencies. It contains 33 scale-based questions with six categories of increasing food frequency, ranging from ‘never’ to ‘several times per day’. The third part concerns nutritional beliefs and consists of 25 questions, each with three possible answers: ‘true’, ‘false’, or ‘difficult to say’. Only ‘true’ answers are scored. The fourth part consists of questions related to lifestyle and personal information. It contains 30 questions concerning the respondent’s demographic data, smoking, preferences related to drinking alcohol, eating out, sleeping, physical activity, subjective assessment of health, nutritional knowledge, and nutritional status. 

The respondents for this study also declared their body mass, height, and waist circumference. This analysis presents 9 variables associated with demographic data. Thirteen questions related to lifestyle were validated. 

Three diet indexes were created based on the 24 items: the Pro-Healthy Diet Index (PHDI) (covering 10 food products: questions no. 23, 25, 31 31–33, 37, 38, 40, and 42–43); the Non-Healthy Diet Index (NHDI) (14 products: questions no. 22, 24, 26–29, 34–36, 44, 46, 51–52, and 54); and the Total-Diet Quality Index (DQI), which was based on the consumption of all products included in the PHDI and NHDI. The total score range is 0–20 points for the PHDI, 0–28 points for the NHDI, and −100–100 for the DQI. All three indexes were created by expressing food frequency as times/day, according to the following formulas:

PHDI: the sum of food frequencies of 10 food groups (times/day);NHDI: the sum of food frequencies of 14 food groups (times/day);DQI: the product of the sum of food frequencies (expressed as times/day) of 24 food groups and weight coefficients.

The sum of consumption frequencies was used to categorise each respondent to one of three intensity groups of nutritional characteristics: low (PHDI: 0.00–6.66; NHID: 0.00–9.33), moderate (PHID: 6.67–13.33; NHID: 9.34–18.66), and high (PHID: 13.34–20.00; NHID: 18.67–28.00). The following score ranges were created for the DQI: −100–−25 indicating high intensity of unhealthy characteristics; −24–24 indicating low intensity of healthy and unhealthy characteristics; and 25–100 indicating high intensity of healthy characteristics. 

For nutritional beliefs, 1 point was awarded for every correct answer to a statement. The total obtainable score was 25 points. Based on the criteria proposed by the authors of the questionnaire, three groups of respondents were distinguished according to their nutritional knowledge: unsatisfactory (0–8 points), satisfactory (9–16 points), and good (17–25 points) [14].

## 3. Statistics

The distributions of the quantitative variables were assessed using the Shapiro–Wilk test. Comparative analyses were conducted using the Mann–Whitney test (for quantitative characteristics) and the chi-squared test (for qualitative characteristics). Reliability was assessed using several statistical procedures. For food frequency, the consumption categories were converted into frequency/day based on the following values: ‘none’ as 0.0, ‘1–3 times per month’ as 0.06, ‘once per week’ as 0.5, ‘several times per week’ as 0.14, ‘once per day’ as 1, and ‘a few times per day’ as 2. Mean values, SD, median, and the interquartile range were calculated for the 28 food products that made up the Pro-Healthy Diet Index, the Non-Healthy Diet Index, and the Total-Diet Quality Index. Differences in food frequency between the test and the retest were assessed using the Wilcoxon matched-pairs test. A cross-classification analysis was performed, and Cohen’s kappa coefficients were determined for the quantitative variables, including food frequency; intensity of the characteristics included in the three nutritional indexes created for this study; and awareness and lifestyle, including the respondents’ physical activity, sleeping time, smoking, and subjectively assessed health. The values of Cohen’s kappa coefficient were classified as follows: 0.00–0.20 indicating no agreement; 0.21–0.39 indicating minimal; 0.40–0.59 indicating weak; 0.60–0.79 indicating moderate; 0.80–0.90 indicating strong; and above 0.90 indicating almost perfect agreement [17]. Cronbach’s alpha was used to assess the internal consistency between diet quality and awareness, where values ranging from 0.7 to 0.9 were considered to be the most desirable [17,18]. 

The minimal sample size was calculated. Based on the kappa coefficient, with a target power of the test of 80% for a minimum acceptable kappa value of 0.40, an expected dropout rate of 10% (in retest), and a significant *p* level of 0.05, the minimum sample size was *n* = 165 [19].

The statistical analysis was performed using the Statistica 13.3 software (Statsoft.pl, Krakow, Poland). The results were considered statistically significant at *p* ≤ 0.05. 

## 4. Results

The mean age of the study participants was 21.31 ± 3.91 years. No significant difference in age was found between the students from Germany and Slovakia. A vast majority of both groups (over 70%) were women. Over 2/3 of the respondents from Germany lived in urban areas, whereas the respondents from Slovakia usually lived in rural areas. The students’ economic situation was predominantly average. However, because the students from Germany assessed their situation as below average more often than their peers from Slovakia, significant differences were also found in the assessment of household economic situation. The German students assessed their economic situation as ‘we live normally’ considerably more often than the Slovakian students; in contrast, the Slovakian students responded ‘we live relatively wealthy’ or ‘very wealthy’ more often than their peers from Germany. The average number of household members was higher among the German students than the Slovakian students. Physical activity and screen time varied drastically in both groups. Over 90% of the German students assessed their activity as average or high, and four out of five used a computer or watched television for a duration between 4 and 10 h per day. The Slovakian students declared moderate activity and usually spent up to 6 h daily using a computer or watching television. Most respondents slept for 7–8 h per day on weekdays. Most respondents also declared that they did not smoke, at the time of the assessment as well as before, and that they were as healthy as their peers (Table 1).

Only three significant differences were observed for the food frequencies of the different products assessed in the test and retest, specifically, for the consumption of milk (German students) and white bread and fried meals (Slovakian students). The food frequencies of other products included in the indexes and the indexes created based on them, i.e., the Pro-Healthy Diet Index, the Non-Healthy Diet Index, and the Total-Diet Quality Index, were similar between the test and retest (Table 2). The frequencies of correct and incorrect responses and the total score obtained in the questions concerning food and nutritional awareness were similar between the test and the retest. The only significant differences in both groups were observed for the statements relating to ‘fruit and vegetables’ and ‘sun exposure increases the synthesis of vitamin D in the human body.’ Furthermore, the students from Germany showed significant test-retest differences in their responses concerning ‘phosphorus as a component of neural tissue’. The Slovakian students showed significant test-retest differences in their responses to the following statements: ‘frequent consumption of grilled meats contributes to the onset of cancer’, ‘carbohydrates should be replaced with simple sugars’, and ‘the ratio of calcium to phosphorus in a healthy diet should be 1:1 (Table 3). The analysis of the remaining questions indicated significant test-retest differences in the declared frequencies of the consumption of sweetened hot beverages and physical activity during leisure time in both groups. In the Slovakian students, differences also emerged for the frequencies of consumption of vegetables and savoury snacks between meals and their preference for grilled meats (Appendix A).

The cross-classification statistics showed agreement for all the variables which were included (or considered) in the analysis. The kappa coefficients ranged from 0.21 for Question No. 60 to 1.0 for Questions No. 52 and 54 among the German students. The range was similar in the Slovakian students: from 0.19 (No. 59) to 0.91 (No. 15). The lowest kappa values among the German students were observed for the questions related to their nutritional knowledge (0.21–0.29), including ‘limiting high-fat foods in everyday diet is protective against cardiovascular diseases’, ‘butter and fortified margarines have a high content of vitamin A and D’, and ‘vegetable oils and olive oil contain a high amount of cholesterol’, as well as for the frequencies of their consumption of vegetable oils, margarines, mixes of butter and margarine, fried foods, and butter. The highest kappa coefficient values were observed for sweetened beverages and alcoholic beverages (1.0). The lowest cross-classification agreement among the Slovakian students was found for their nutritional beliefs related to the statement, ‘a high intake of salt protects against hypertension’. The highest cross-classification values were observed for two variables relating to lifestyle and personal data, including type of alcohol usually consumed and currently smoking habit, and for one variable relating to dietary habits, namely consumption of sweetened hot beverages (Table 4). 

The Cronbach’s ɑ values ranged from 0.58 to 0.78, depending on the phase of testing and the group of questions. The internal consistency of the questions related to the German students’ knowledge in the first phase was moderate: 0.58 (mean intra-correlation was 0.05). However, a higher internal consistency was observed in the retest: 0.75 (mean intra-correlation was 0.11). The corresponding values among the Slovakian students equalled to 0.68 (mean intra-class correlation, *r* = 0.08) and 0.77 (*r* = 0.13), respectively. For the questions related to dietary habits, the internal consistency among the German students equalled to 0.74 (*r* = 0.08) in both the test and the retest. The corresponding values for the Slovakian students equalled to 0.75 (*r* = 0.09) and 0.78 (*r* = 0.10). 

## 5. Discussions

This study is the first one to use the KomPAN questionnaire to assess the dietary habits, food frequency, and nutritional knowledge among university students in two countries neighbouring Poland [20]. Previously, the KomPAN was assessed for its reproducibility and was validated with respect to both its self-administered (SA-Q) and interviewer-administered (IA-Q) versions in Poland among a group of 831 healthy individuals aged 15–65 years; the KomPAN was also validated with respect to the SA-Q version only among a group of 148 patients with metabolic disorders, inflammatory bowel disease, hypertension, and diabetes mellitus type 1 [15]. Even though an English version of the KomPAN is available, the questionnaire has not been assessed for its reproducibility or validated in any other European country. 

Tests to confirm the reliability and reproducibility of self-administered food frequency questionnaires and nutritional knowledge questionnaires have been conducted by numerous centres in Poland [15,21,22,23] and abroad [24]. Several previous studies have focused on assessing the nutritional knowledge among university students [25,26,27,28]. An article by Cade et al. (2004) showed that over 70% of all studies concerning food frequency were based on SA-Q versions [19]. SA-Qs are also the most frequently used tools in online studies [29]. 

In this study, different sections of the KomPAN yielded different values of Cronbach’s ɑ. The Dietary Habits and Food Frequency sections showed a high internal consistency, especially in the retest. The obtained results were generally consistent with the widely accepted Cronbach’s ɑ scoring and were similar to those in other validated nutritional questionnaires (α > 0.7), but were lower (in this test) than those obtained in general nutritional knowledge tests [23,26,30,31]. Low values of Cronbach’s ɑ were obtained in the Food Choices section of the Australian Nutrition Knowledge Questionnaire in a study conducted among Australian university students [30]. An earlier study based on the same questionnaire also yielded low values of ɑ for two sections: Dietary Recommendations (0.53) and Choosing Everyday Foods (0.55) [31]. A Cronbach’s ɑ below 0.5 indicates that the correlation between the items included in a section is weak; therefore, the given set of questions should not be used on its own, but rather should be used together with the other sections of the questionnaire [30].

An analysis of the temporal consistency of the SA-Q in the students from Germany and Slovakia demonstrated a very good or good reproducibility of the results related to dietary habits [32]. A cross-classification analysis only found a below-acceptable consistency (<50%) for a few questions. For the vast majority of the questions, the kappa coefficients were higher than the acceptable value of 0.40 for the nutrients of interest in epidemiological studies [32]. Only 13 nutrition-related questions among the German students showed a below-acceptable kappa. A low consistency was also observed in a study on the intake of macro- and micronutrients in Lebanese elders [33], as well as in a study assessing food frequency among Romanian adults [34]. A Polish study conducted by Kowalkowska et al. (2012) with a SA-Q found below-acceptable values of kappa for a third of the 33 products included in the nutritional indexes for outpatients [15]. There are several possible explanations for this. First and foremost, the respondents might find it more difficult to recall and declare the food frequencies for products that are consumed more than 1–3 times per month than for those that are consumed rarely or not at all. Secondly, a description of a product that lists too many meals as examples of its consumption frequency makes it even more difficult for the subjects to choose a response [15]. Another explanation is related to the fact that university students, as a group, have yet to develop their dietary habits and are unlikely to change them within a short period of time. This is likely associated with their irregular lifestyle that involves eating a varying number of main meals or skipping meals entirely, especially breakfast [35]. The assessment of the consumption of 24 food products proposed in the KomPAN was the basis for a separate assessment of the quality of healthy, unhealthy, and general eating habits of the students from Germany and Slovakia. In both the test and the retest, a cross-classification analysis showed a high internal consistency and a good or acceptable reproducibility. The results of this study are similar to those obtained by Kowalkowska et al. among a group of healthy individuals using the SA-Q version of the KomPAN [15]. However, they are also much higher than the results obtained in other studies [36,37]. For example, an assessment of the reliability of the New Nordic Diet Score yielded lower index values for the cross-classification analysis (69%) and kappa (0.5) [36]. Another study, conducted among Portuguese youth using the Mediterranean Diet Quality Index, also demonstrated a lower reproducibility [37]. It should be noted that all of the aforementioned studies maintained a similar, two-week interval between the test and the retest. 

This study demonstrated a good or moderate consistency with respect to the questions related to nutritional knowledge and beliefs (number of questions: 25), the total score obtained by the participants (objective knowledge), and the subjectively assessed knowledge (*N* = 93). Slightly higher kappa values were observed for the subjective assessment than for the objective assessment of knowledge. The Slovakian students showed a higher temporal consistency than the German students, in whom the kappa values for individual statements were lower than 0.30. In a study conducted by Kowalkowska et al., the total level of nutritional knowledge amounted to 0.71 (kappa) in healthy individuals and was considerably lower in outpatients (0.46) [15]. Additionally, in a study conducted among residents of Quebec, an assessment of the test-retest reliability demonstrated a significant association for the total knowledge score at *r* = 0.59 [38]. In a study conducted among Danish women, the deattenuated Pearson coefficients for different food groups ranged from 0.25 for sugar to 0.75 for fish [39]. Conversely, Fallaize et al. assessed food frequency using an online version of the SA-Q and obtained a much broader range of Spearman correlation coefficients: 0.11–0.73 [40]. However, it is worth pointing out that this study and the study conducted by Kowalkowska et al. (2018) were the only ones that used an identical procedure to calculate reliability. In the other studies, the temporal consistency was based on correlation coefficients [15]. It goes without saying that even high values of correlation coefficients do not necessarily indicate a test-retest consistency, because the correlations are primarily based on the strength of the association between both assessments and do not display mutual consistency between the results. The kappa value is a more accurate indicator of the mutual consistency between the results. Consequently, it is difficult to state which studies had the highest consistency. 

It seems that the two-week interval between the test and the retest might have helped the participants to respond correctly to the questions. According to di Lorio, even an interval as short as two weeks between assessments may affect the level of nutritional knowledge by encouraging information flow [41]. 

Some topics related to nutritional knowledge proved to be difficult for the students. Furthermore, some of the responses were classified as incorrect more often in the retest than in the test. It is worth emphasising that as many as two out of the three categories available as responses were considered incorrect, which is why the students who did not know the answer to a given question more often selected a different category that was also considered incorrect. A few studies have obtained considerably different results between the test and the retest [15,26,31]. 

The analysis of the results related to lifestyle showed that the questions with fewer categories of responses (e.g., smoking) demonstrated a much higher reproducibility among both groups of students than the questions with more categories (e.g., eating out or time spent sleeping, watching TV, or time spent on a computer). Consequently, the obtained results confirm the hypothesis that simple questions are more reliable, as opposed to questions with many categories of responses. A similar relationship has been observed in studies conducted based on the SA-Q version of the KomPAN [15]. This research has certain limitations. First and foremost, extensive questionnaires are more likely to be filled in incompletely than short ones. It seems that, in the proposed questionnaire, the overall indicators of diet quality allow for a slight modification for the purpose of future studies, namely the number of questions related to the frequency of consumption can be reduced from 33 to 24. In some cases, this modification may help not only to reduce the cost of preparing the questionnaire and shorten the research time, but also to collect a higher percentage of correctly and completely filled-in questionnaires [42]. It should also be mentioned that, because the KomPAN does not specify the size of a portion, it is impossible to obtain data related to the caloric and nutritional values of the respondents’ standard diet [15]. In the case of the responses concerning some food groups and beliefs about food and nutrition, especially those provided by the students from Germany, the results should be interpreted with caution due to the low reproducibility coefficients. Furthermore, this study was conducted during the SARS-CoV-2 pandemic, which might have affected the outcomes. For example, the pandemic enforced limitations on activities in schools, cafeterias, restaurants, eateries, etc. It also likely forced individuals to change how often they went shopping for food, as shopping was considered to bear the risk of a COVID-19 infection. As a result, people more often bought products with a long expiration date [43]. Significant differences were observed in the economic situation of the household between the Slovakian and German students. However, the fact that the Slovakian students rated their household situation higher (“relatively wealthy” or “very wealthy”) than the German students did not necessarily mean that their living conditions were actually better than their German peers, as shown by some macroeconomic data based on indicators obtained in Slovakia and Germany [44].

The strengths of this study should also be mentioned. A sample that included a relatively large group of university students of the same age is a definite strength, as it allows for validation among a group of students that is uniform in terms of age. Earlier studies presenting the validation of questionnaires similar to the KomPAN were conducted among similar-sized or smaller samples of individuals of different ages [21,45]. The most recent validation of the current version of the KomPAN included a large sample. However, it focused on comparing the applicability of two versions: the SA-Q among a group of healthy individuals and outpatients with chronic diseases, and the IA-Q among a group of healthy individuals. Furthermore, that study only concerned the Polish population [15]. Our study also validated three diet indices: the Pro-Healthy Diet Index, the Non-healthy Diet Index, and the Total-Diet Quality Index, the last of which we validated for the first time.

The reproducibility of the KomPAN among an age-limited group of university students and its acceptance in full will allow researchers to analyse and compare the dietary habits of residents of Germany and Slovakia in a wider scope in future studies. This will expand the possibilities for such studies across Europe and lay the foundation for the development of a unified research tool for the assessment of nutritional habits among culturally similar groups and/or countries. It should also be emphasised that our reliability and reproducibility assessment incorporated several methods of statistical analysis that have been used in other studies in order to improve the robustness of the results and conclusions [15,46,47]. Furthermore, the analysis was performed based on food frequencies, which helped us systematise the analysis of food consumption [15,37,46,48].

## 6. Conclusions

The conducted validation of the KomPAN questionnaire demonstrated good or moderate reproducibility of the results related to dietary habits, lifestyle, and nutritional knowledge, which were obtained using the German and Slovakian versions of the questionnaire. This allows the questionnaire to be recommended as an accurate and inexpensive self-assessment questionnaire (SA-Q) that can provide reliable information about lifestyles and frequencies of consumption of usual foods in adults. The questionnaire could be used in large-scale epidemiological studies conducted in European countries in order to assess and compare the dietary habits of their residents. The obtained results also encourage further research on the development of a unified tool for the assessment of nutritional knowledge and lifestyles in other countries.

Furthermore, the number of proposed categories was demonstrated to affect the test-retest reproducibility. In particular, questions with only two answers are more reproducible than those with more than two answers.

## Figures and Tables

**Figure 1 nutrients-14-04893-f001:**
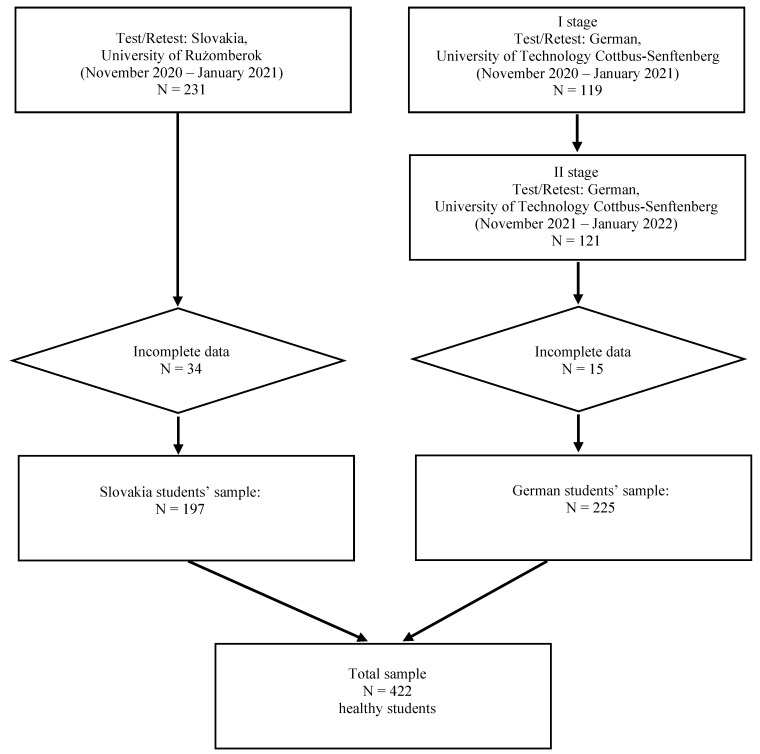
Study design and sample size.

**Table 1 nutrients-14-04893-t001:** Characteristics of the study participants.

Variable	Germany (*N* = 225)		Slovakia (*N* = 197)		*p*
	*n*	%	*n*	%	
Sex					
Women	166	73.8	153	77.7	ns
Men	59	26.2	44	22.3	
Age M (SD)	21.6	4.0	21.0	3.9	ns
Place of residence					
Village	62	27.6	122	61.9	<0.001
Small town (<20,000 inhabitants)	61	27.1	32	16.2	
Town (20,000–100,000 inhabitants)	66	29.3	35	17.8	
City (>100,000 inhabitants)	36	16.0	8	4.1	
Economic situation of the family					
Below average	61	27.1	25	12.7	<0.001
Average	150	66.7	166	84.3	
Above average	13	5.8	6	3.0	
Economic situation of the household					
We live modestly or very modestly	15	6.7	1	0.5	<0.001
We live normally	114	50.7	8	4.1	
We live relatively affluently	77	34.2	81	41.1	
We live very affluently	18	8.0	96	48.7	
Number of persons in the family: Me(IQR)	2(3)		5(2)		<0.001
Lifestyle					
Physical activity during school or work ^1^					
Low	122	54.2	51	25.9	<0.001
Average	95	42.2	128	65.0	
High	8	3.6	18	9.1	
Physical activity during leisure time ^2^					
Low	16	7.1	42	21.3	<0.001
Average	106	47.1	129	65.5	
High	103	45.8	26	13.2	
Screen time					
Less than 2 h	16	7.11	63	32.0	<0.001
From 2 to almost 4 h	59	26.2	74	37.6	
From 4 to almost 6 h	80	35.6	41	20.8	
From 6 to almost 8 h	55	24.4	16	8.1	
From 8 to almost 10 h	8	3.6	1	0.5	
More than 10 h	7	3.1	2	1.0	
Sleep time—weekdays					
6 or less hours/day	44	19.6	72	36.5	<0.001
More than 6 but less than 9 h/day	174	77.3	117	59.4	
9 or more hours/day	7	3.1	8	4.1	
Sleep time—weekends					
6 or less hours/day	10	4.4	24	12.2	0.05
More than 6 but less than 9 h/day	129	57.3	105	53.3	
9 or more hours/day	86	38.2	68	34.5	
Currently smoking cigarettes, pipe, or tobacco					
No	179	79.6	153	77.7	ns
Yes	49	20.4	44	22.3	
Smoked cigarettes in the past					
No	142	63.1	125	63.5	ns
Yes	83	36.9	72	36.5	
Health status in comparison to other people your age					
Worse than others	21	9.3	27	13.7	ns
The same as others	164	72.9	140	71.1	
Better than others	40	17.8	30	15.2	

M(SD): mean (standard deviation); Me(IQR): median (interquartile range); ns: not statistically significant; *p*: significance level: 0.05, 0.01, 0.001, and <0.001.^1^ Physical activity during school or work: low physical activity—over 70% sitting time; average physical activity—about 50% sitting time and about 50% time spent on physical activity; and high physical activity—about 70% % time spent on physical activity or intense physical labour. ^2^ Physical activity during leisure time: low physical activity—a majority of time spent sitting, watching TV, reading the press or books, doing light housework, and walking 1–2 h per week; average physical activity—walking, cycling, aerobics, gardening, and other light physical activity 2–3 h per week; and high physical activity—cycling, running, gardening, and other recreational sports activities requiring physical effort performed for over 3 h per week.

**Table 2 nutrients-14-04893-t002:** Study participants’ dietary habits in the test and the retest.

Question	Products	Germany				*p*	Slovakia				*p*
		Test		Retest			Test		Retest		
		M(SD)	Me(IQR)	M(SD)	Me(IQR)		M(SD)	Me(IQR)	M(SD)	Me(IQR)	
23	Wholemeal (brown) bread/bread rolls	0.5 (0.4)	0.5 (0.4)	0.4 (0.4)	0.5 (0.4)	ns	0.3 (0.4)	0.1 (0.4)	0.3 (0.4)	0.1 (0.4)	ns
25	Buckwheat, oats, wholegrain pasta, or other coarse-ground groats	0.3 (0.4)	0.1 (0.4)	0.3 (0.4)	0.1 (0.4)	ns	0.3 (0.4)	0.1 (0.4)	0.3 (0.4)	0.1 (0.4)	ns
31	Milk (including flavoured milk, hot chocolate, and lattes)	0.5 (0.5)	0.5 (0.4)	0.4 (0.5)	0.1 (0.4)	0.05	0.6 (0.6)	0.5 (0.9)	0.6 (0.6)	0.5 (0.4)	ns
32	Fermented milk beverages, e.g., yoghurts, kefir (natural or flavoured)	0.3 (0.3)	0.1 (0.4)	0.3 (0.4)	0.1 (0.4)	ns	0.5 (0.4)	0.5 (0.4)	0.4 (0.4)	0.5 (0.4)	ns
33	Fresh cheese curd products, e.g., cottage cheese, homogenised cheese, fromage frais	0.2 (0.3)	0.1 (0.0)	0.2 (0.3)	0.1 (0.0)	ns	0.2 (0.3)	0.1 (0.0)	0.2 (0.2)	0.1 (0.4)	ns
37	White meat, e.g., chicken, turkey, rabbit	0.3 (0.2)	0.1 (0.4)	0.2 (0.2)	0.1 (0.4)	ns	0.4 (0.4)	0.5 (0.4)	0.4 (0.4)	0.5 (0.4)	ns
38	Fish	0.2 (0.2)	0.1 (0.1)	0.1 (0.20)	0.1 (0.0)	ns	0.1 (0.1)	0.1 (0.0)	0.1 (0.2)	0.1 (0.04)	ns
40	Pulse-based foods, e.g., foods made from beans, peas, soybeans, and lentils	0.2 (0.3)	0.1 (0.4)	0.2 (0.3)	0.1 (0.4)	ns	0.1 (0.14)	0.1 (0.08)	0.1 (0.13)	0.1 (0.0)	ns
42	Fruits	1.0 (0.7)	1.00 (1.5)	1.01 (0.63)	1.00 (0.5)	ns	0.87 (0.62)	0.5 (0.5)	0.9 (0.6)	0.5 (0.5)	ns
43	Vegetables	1.0 (0.63)	1.0 (0.5)	1.0 (0.6)	1.00 (0.5)	ns	0.8 (0.6)	0.5 (0.5)	0.8 (0.6)	0.5 (0.5)	ns
22	Wheat bread, rye bread, wheat/rye bread, toast bread, and bread rolls	0.5 (0.5)	0.5 (0.4)	0.4 (0.35)	0.5 (0.4)	ns	0.7 (0.6)	0.5 (0.9)	06 (0.5)	0.5 (0.9)	0.01
24	White rice, white pasta, and fine-ground groats, e.g., semolina, couscous	0.4 (0.3)	0.5 (0.4)	0.4 (0.30)	0.5 (0.4)	ns	0.3 (0.3)	0.1 (0.4)	0.3 (0.3)	0.1 (0.4)	ns
26	Fast foods, e.g., potato chips, hamburgers, pizza, hot dogs	0.1 (0.1)	0.1 (0.0)	0.1 (0.1)	0.1 (0.0)	ns	0.1 (0.23)	0.1 (0.0)	0.1 (0.20)	0.1 (0.0)	ns
27	Fried foods (e.g., meat or flour-based foods such as dumplings and pancakes)	0.3 (0.3)	0.1 (0.4)	0.3 (0.2)	0.1 (0.4)	ns	0.2 (0.26)	0.1 (0.2)	0.2 (0.2)	0.1 (0.0)	0.01
28	Butter as a bread spread or as an addition to your meals (for frying, baking, etc).	0.2 (0.3)	0.1 (0.4)	0.2 (0.35)	0.1 (0.0)	ns	0.4 (0.5)	0.5 (0.4)	0.4 (0.45)	0.5 (0.4)	ns
29	Lard as a bread spread or as an addition to your meals (for frying, baking, etc).	0.02 (0.10)	0.0 (0.0)	0.02 (0.09)	0.0 (0.0)	ns	0.1 (0.2)	0.1 (0.1)	0.1 (0.3)	0.1 (0.1)	ns
34	Cheese (including processed cheese and blue cheese)	0.5 (0.4)	0.5 (0.4)	0.5 (0.4)	0.5 (0.4)	ns	0.2 (0.3)	0.1 (0.1)	0.2 (0.34)	0.1 (0.4)	ns
35	Cold meats, smoked sausages, and hot dogs	0.4 (0.4)	0.5 (0.4)	0.4 (0.4)	0.5 (0.4)	ns	0.2 (0.2)	0.1 (0.1)	0.2 (0.21)	0.1 (0.0)	ns
36	Red meats, e.g., pork, beef, veal, mutton, lamb, game	0.2 (0.2)	0.1 (0.1)	0.2 (0.20)	0.1 (0.1)	ns	0.2 (0.3)	0.1 (0.4)	0.2 (0.2)	0.1 (0.4)	ns
44	Sweets, e.g., confectionary, biscuits, cakes, chocolate bars, cereal bars	0.6 (0.5)	0.5 (0.4)	0.6 (0.50)	0.5 (0.4)	ns	0.7 (0.6)	0.5 (0.7)	0.6 (0.7)	0.5 (0.9)	ns
46	Tinned (jar) meats	0.02 (0.1)	0.0 (0.0)	0.02 (0.1)	0.0 (0.0)	ns	0.0 (0.11)	0.0 (0.1)	0.1 (0.14)	0.0 (0.1)	ns
51	Sweetened carbonated or still beverages, such as Coca-Cola, Pepsi, Sprite, Fanta, and lemonade	0.2 (0.3)	0.1 (0.0)	0.2 (0.25)	0.1 (0.0)	ns	0.2 (0.37)	0.1 (0.0)	0.2 (0.4)	0.1 (0.2)	ns
52	Energy drinks, such as Red Bull, Monster, Rockstar, or others	0.1 (0.2)	0.0 (0.1)	0.1 (0.2)	0.0 (0.1)	ns	0.1 (0.19)	0.0 (0.1)	0.05 (0.13)	0.00 (0.1)	ns
54	Alcoholic beverages	0.2 (0.2)	0.1 (0.0)	0.1 (0.17)	0.1 (0.0)	ns	0.1 (0.20)	0.1 (0.0)	0.1 (0.2)	0.1 (0.0)	ns
Pro-Healthy Diet Index (PHDI) (sum of frequency/day)		4.4 (1.9)	4.2 (2.5)	4.2 (1.89)	3.9 (2.5)	ns	4.1 (1.94)	3.9 (2.5)	4.0 (2.0)	3.7 (13.0)	ns
Intensity of the characteristics *N*; %	Low	223	99.1	224	99.6	ns	223	99.1	224	99.6	ns
	Moderate	2	0.9	1	0.4		2	0.9	1	0.4	
Non-Healthy Diet Index (NHDI) (sum of frequency/day)		3.6 (1.8)	3.2 (2.1)	3.4 (1.6)	3.2 (2.1)	ns	3.7 (1.5)	3.3 (2.4)	3.4 (1.77)	3.2 (2.5)	ns
Intensity of the characteristics *N*; %	Low	198	88.0	204	90.7	ns	67	34.01	67	34.01	ns
	Moderate	27	12.0	21	9.3		92	46.69	104	52.8	
	High	0	0.0	0	0.0		38	19.30	26	13.20	
Total-Diet Quality Index (points)		9.2 (10.59)	8.5 (13.1)	8.8 (9.81)	7.2 (13.2)	ns	7.4 (10.98)	6.0 (13.4)	7.7 (11.19)	6.4 (13.0)	ns
Intensity of the characteristics *N*; %	Low intensity of non-healthy and healthy characteristics	211	93.8	216	96.0	ns	9	4.6	13	6.61	ns
	High intensity of healthy characteristics	14	6.2	9	4.0		188	95.4	184	93.4	

M(SD): mean (standard deviation); Me(IQR): median (interquartile range); ns: not statistically significant; *p*: significance level: 0.05, 0.01, 0.001, and <0.001.

**Table 3 nutrients-14-04893-t003:** Study participants’ beliefs about food and nutrition (%).

Question	Knowledge	Germany				*p*	Slovakia				*p*
		Test		Retest			Test		Retest		
		Correct	False	Correct	False		Correct	False	Correct	False	
55	It is enough to eat wholegrains/cereals once a day.	8.7	91.3	12.0	88.0	ns	16.2	83.8	19.8	80.2	ns
56	Only children and adolescents should drink milk.	71.0	29.0	72.1	27.9	ns	81.2	18.8	83.2	16.8	ns
57	Fruits and/or vegetables should be consumed with every meal.	80.9	19.1	82.0	18.0	ns	66.5	66 33.5	138 70.1	59 29.9	ns
58	Consumption of mouldy bread can result in food poisoning caused by Salmonella.	49.7	50.3	48.6	51.4	ns	40.6	59.4	39.6	60.4	ns
59	A high intake of salt protects against hypertension.	82.0	18.0	76.0	24.0	ns	89.3	13.7	81.2	18.8	ns
60	Limiting high-fat foods in everyday diet is protective against cardiovascular diseases.	89.6	10.4	89.1	10.9	ns	88.3	11.7	84.3	15.7	ns
61	Frequent consumption of oily fish contributes to atherosclerosis.	16.9	83.1	21.9	78.1	ns	34.5	65.5	36.0	64.0	ns
62	Frequent consumption of grilled meats contributes to the onset of cancer.	46.4	53.6	50.3	78.1	ns	23.9	76.1	30.5	69.5	0.05
63	A vegetarian diet increases the risk of anaemia.	16.9	83.1	18.0	82.0	ns	66.0	34.0	71.1	28.9	ns
64	Bio-yoghurts contain beneficial gut bacteria.	57.4	42.6	61.2	38.8	ns	61.9	38.1	64.5	35.5	ns
65	Vegetable oils and olive oil contain a high amount of cholesterol.	26.8	73.2	22.4	77.6	ns	34.0	66.0	28.9	71.1	ns
66	Wholemeal bread has more fibre than white bread.	84.7	15.3	80.3	19.7	ns	77.7	22.3	72.6	27.4	ns
67	Fruits and vegetables are a source of ‘empty calories’.	55.2	44.8	43.7	56.3	0.01	37.6	62.4	28.9	71.1	0.05
68	Butter and fortified margarines have a high content of vitamins A and D.	16.4	83.6	19.7	80.3	ns	29.9	70.1	34.0	66.0	ns
69	Cheese is a better source of calcium than cottage cheese.	25.7	74.3	24.6	75.4	ns	13.7	86.3	17.8	82.2	ns
70	Offal has high amounts of ‘bad’ cholesterol—LDL.	12.6	87.4	10.9	89.1	ns	18.3	86.7	20.3	79.7	ns
71	In a healthy diet, complex carbohydrates should be replaced with simple sugars.	61.7	38.3	54.6	45.4	ns	23.4	76.6	15.7	84.3	0.01
72	In a balanced diet, proteins should be the main source of energy.	26.2	73.8	27.3	72.7	ns	14.2	85.8	16.8	83.2	ns
73	Inadequate intakes of vitamin PP can cause skin inflammation and diarrhoea.	25.1	74.9	28.4	71.6	ns	25.9	71.1	27.9	72.1	ns
74	Sun exposure increases the synthesis of vitamin D in the human body.	95.1	4.9	88.0	12.0	0.001	90.4	9.6	83.2	16.8	0.01
75	Phosphorus is a component of the neural tissue.	28.4	71.6	37.2	62.8	0.05	31.0	69.0	35.0	65.0	ns
76	The ratio of calcium to phosphorus in a healthy diet should be 1:1.	8.2	91.8	8.7	91.3	ns	42.6	57.4	35.5	64.5	0.01
77	Consumption of fruits with a high content of vitamin C increases the bioavailability of iron.	58.5	41.5	57.9	42.1	ns	49.7	50.3	50.3	49.7	ns
78	Starting the cooking of vegetables in cold water helps to preserve the nutrients.	32.2	67.8	25.7	74.3	ns	24.9	79.7	16.8	83.2	ns
79	Sweets and animal fats are particularly high nutrient-dense foods.	47.0	53.0	43.2	56.8	ns	24.9	75.1	22.3	77.7	ns
Sum of nutritional knowledge: M(SD) and Me(IQR)		11.23 (3.18)	11.0 (4.0)	11.04 (3.52)	11.0 (4.0)	ns	10.99 (3.24)	11.0 (4.0)	10.86 (3.52)	11.0 (5.0)	ns

M(SD): mean (standard deviation); Me(IQR): median (interquartile range); ns: not statistically significant; *p*: significance level: 0.05, 0.01, 0.001, and <0.001.

**Table 4 nutrients-14-04893-t004:** Consistency coefficients for the test and retest in both groups of students.

Question	No. of Categories	Slovakia		Germany	
		Consistency (%) (the Same Category)	Kappa	Consistency (%) (the Same Category)	Kappa
Dietary habits					
Number of meals	5	80.31	0.72	72.97	0.53
Consume meals at regular times	3	78.87	0.61	72.65	0.53
Snacking between meals	6	58.55	0.47	48.89	0.32
Fruits	2	85.57	0.58	86.67	0.51
Vegetables	2	74.74	0.50	73.78	0.48
Unsweetened dairy beverages and desserts	2	80.93	0.58	71.56	0.37
Sweetened dairy beverages and desserts	2	78.87	0.45	82.67	0.43
Sweet snacks	2	85.05	0.60	78.22	0.51
Savoury snacks	2	88.66	0.66	79.11	0.58
Nuts, almonds, and seeds	2	89.18	0.73	73.78	0.48
Type of milk	3	89.01	0.78	89.53	0.80
Prepared meat: boiled	2	89.19	0.69	80.27	0.54
Prepared meat: stewed	2	86.53	0.70	92.83	0.46
Prepared meat: grilled	2	88.14	0.74	77.58	0.54
Prepared meat: roasted	2	91.75	0.81	76.58	0.44
Prepared meat: fried	2	89.18	0.77	89.19	0.68
I don’t eat meat	2	98.45	0.66	97.31	0.90
Fat for spreading	7	89.64	0.75	63.94	0.50
Fat for frying	6	89.12	0.77	77.00	0.57
Sweetened hot beverages	4	93.88	0.91	86.61	0.66
Adding salt to ready meals	3	88.72	0.80	79.91	0.56
Type of water usually drink: I don’t drink water	2	99.45	0.80	89.78	0.50
Type of water usually drink: still water	2	93.88	0.86	92.89	0.79
Type of water usually drink: sparkling water	2	88.27	0.73	89.78	0.78
Type of water usually drink: flavoured water	2	88.76	0.72	92.00	0.63
Food frequency—items not included in the nutritional indexes					
Vegetable oils, margarines, or mixes of butter and margarine as a bread spread	6	55.84	0.41	46.67	0.27
Eggs	6	71.07	0.61	60.89	0.47
Potato	6	70.56	0.52	57.78	0.40
Instant soups or ready-made soups	6	68.02	0.56	65.33	0.45
Tinned (jar) vegetables	6	66.50	0.46	57.33	0.34
Fruit juices	6	64.47	0.52	57.78	0.41
Vegetable juices or fruit and vegetable juices	6	60.91	0.46	60.44	0.37
Sweetened hot beverages	6	52.79	0.41	46.22	0.31
Water	6	73.10	0.53	88.00	0.48
Lifestyle					
Eating out	6	73.80	0.63	62.39	0.41
Type of alcohol usually drink	5	96.82	0.94	83.14	0.76
Currently smoke	2	97.44	0.93	91.44	0.73
Smoke in the past	2	94.30	0.88	87.89	0.74
Sleep a day during weekdays	3	90.10	0.83	82.88	0.55
Sleep a day during weekends	3	87.11	0.78	79.37	0.61
Screen time	5	76.41	0.68	51.35	0.34
Physical activity in school/work	4	85.42	0.71	68.02	0.40
Physical activity in leisure time	3	85.94	0.71	72.65	0.52
Health status in comparison to other people your age	3	89.64	0.77	78.03	0.49
Knowledge about nutrition	4	79.17	0.68	71.75	0.57
Description of diet	4	91.19	0.82	82.35	0.61
Differences between diet during weekdays compared to weekends	3	72.77	0.56	63.34	0.31
Food frequencies—items included in the nutritional indexes					
Wholemeal (brown) bread/bread rolls	6	62.76	0.53	55.11	0.71
Buckwheat, oats, and wholegrain pasta	6	69.90	0.61	56.44	0.44
Milk	6	64.80	0.54	58.67	0.69
Fermented milk beverages	6	58.67	0.45	50.22	0.69
Fresh cheese curd products	6	70.92	0.58	56.00	0.41
White meat	6	70.92	0.56	62.67	0.50
Fish	6	76.02	0.63	72.00	0.51
Pulse-based foods	6	75.00	0.60	51.56	0.32
Fruits	6	66.33	0.54	58.22	0.42
Vegetables	6	66.84	0.54	56.89	0.40
White bread and bakery products	6	64.28	0.54	49.78	0.32
White rice, white pasta, and fine-ground groats	6	67.34	0.54	56.44	0.59
Fast foods	6	73.98	0.53	66.22	0.28
Fried foods	6	68.37	0.54	51.11	0.27
Butter	6	58.16	0.46	45.33	0.28
Lard	6	62.24	0.46	82.22	0.37
Cheese	6	60.20	0.47	52.44	0.60
Cold meats, smoked sausages, and hot dogs	6	67.35	0.53	64.44	0.53
Red meat	6	68.88	0.57	64.00	0.52
Sweets	6	67.86	0.57	51.11	0.51
Tinned (jar) meats	6	81.12	0.59	80.44	0.42
Sweetened carbonated or still beverages	6	67.35	0.56	65.33	0.51
Energy drinks	6	89.80	0.80	77.33	0.66
Alcohol	6	83.16	0.71	68.00	0.53
PHDI ^1^	3	91.33	0.60	88.45	0.41
NHDI ^2^	3	95.45	0.59	99.55	0.66
DQI ^3^	3	95.92	0.62	94.22	0.41
Nutritional beliefs					
Only children and adolescents should drink milk.	3	70.05	0.42	67.21	0.31
Fruits and/or vegetables should be consumed with every meal.	3	94.42	0.82	74.86	0.44
Consumption of mouldy bread can result in food poisoning caused by Salmonella.	3	79.70	0.59	80.33	0.38
A high intake of salt protects against hypertension.	3	77.66	0.66	66.12	0.46
Limiting high-fat foods in everyday diet is protective against cardiovascular diseases.	3	84.26	0.50	77.60	0.36
Frequent consumption of oily fish contributes to atherosclerosis.	3	84.71	0.38	84.70	0.21
Frequent consumption of grilled meats contributes to the onset of cancer.	3	69.54	0.51	66.67	0.42
A vegetarian diet increases the risk of anaemia.	3	68.53	0.52	69.40	0.51
Bio-yoghurts contain beneficial gut bacteria.	3	82.23	0.63	63.93	0.41
Vegetable oils and olive oil contain a high amount of cholesterol.	3	73.60	0.50	67.21	0.40
Wholemeal bread has more fibre than white bread.	3	67.00	0.50	57.38	0.29
Fruits and vegetables are a source of ‘empty calories’.	3	77.66	0.44	78.69	0.30
Butter and fortified margarines have a high content of vitamins A and D.	3	71.57	0.57	56.28	0.31
Cheese is a better source of calcium than cottage cheese.	3	86.80	0.80	55.74	0.28
Offal has high amounts of ‘bad’ cholesterol—LDL.	3	72.59	0.50	60.11	0.36
In a healthy diet, complex carbohydrates should be replaced with simple sugars.	3	72.59	0.55	73.22	0.42
In a balanced diet, proteins should be the main source of energy.	3	72.08	0.56	68.85	0.45
Inadequate intakes of vitamin PP can cause skin inflammation and diarrhoea.	3	82.74	0.61	60.11	0.36
Sun exposure increases the synthesis of vitamin D in the human body.	3	77.66	0.61	68.85	0.38
Phosphorus is a component of the neural tissue.	3	87.82	0.49	89.07	0.34
The ratio of calcium to phosphorus in a healthy diet should be 1:1.	3	73.09	0.54	63.39	0.38
Consumption of fruits with a high content of vitamin C increases the bioavailability of iron.	3	73.60	0.53	79.23	0.39
Starting the cooking of vegetables in cold water helps to preserve the nutrients.	3	70.56	0.50	63.39	0.32
Only children and adolescents should drink milk.	3	72.59	0.57	58.47	0.37
Sweets and animal fats are particularly high nutrient dense foods.	3	69.03	0.52	66.12	0.45
Level of nutritional knowledge	3	78.17	0.48	75.96	0.43

^1^ PHDI: Pro-Healthy Diet Index; ^2^ NHDI: Non-Healthy Diet Index; ^3^ DQI: Total-Diet Quality Index.

## Data Availability

The data presented in this study are available from the corresponding author upon request.

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
