# Peer review of "Reproducibility of the German and Slovakian Versions of the Dietary Habits and Nutrition Beliefs Questionnaire (KomPAN)"

_nutrients, 2022, doi:10.3390/nu14224893_

Round 1

Reviewer 1 Report

Manuscript ID: nutrients- 2012028

Authors: Elżbieta Cieśla et al.

Title: "Reproducibility of the German and Slovakian versions of the Dietary Habits and Nutrition Beliefs Questionnaire (KomPAN)"

The authors of the present study tried to assess the reproducibility of the German and Slovakian language versions of the Dietary Habits and Nutrition Beliefs Questionnaire (KomPAN), a previously developed in Poland.

The following comments should be considered:

Comments:

1.     Please clarify the abbreviations used in the Tables (e.g., Table 1: "Mo")

2.     It would be helpful if the authors could add a flow chart and summarize the inclusion and exclusion criteria of the study.

3.     Did the authors perform a power and sample size calculation?

4.     It should be noted that the study was conducted among young (the mean age of the participants is 21.0 and 21.6 in the "Slovakia" and "German" cohorts, respectively) and "healthy" adults (probably without chronic or profound metabolic disorders). In addition, the vast majority of the participants in the Slovakia cohort (approximately 90%) live relatively or very affluently. Therefore, the above might constitute a potential limitation of the study regarding the generalizability of its findings that should be acknowledged.

Author Response

Dear Revier,

Thank you for taking the time to review our article and for your important feedback, which will help to improve the scientific value of the article.  Below are our responses to each remark.

Please clarify the abbreviations used in the Tables (e.g., Table 1: "Mo"):

  • In Lines: 206, 216, 240. The following abbreviations are now explained below the table: Me(q1-q3), Mo(minimum – maximum); X(SD), ns; p;

It would be helpful if the authors could add a flow chart and summarize the inclusion and exclusion criteria of the study

  • A flowchart has been added.

Did the authors perform a power and sample size calculation?

Determining the sample size is a complex issue in the case of validating questionnaires that use percentage frequencies and mean values and/or medians. Validation analysis requires multiple statistical tests for repeated measurements (etc.). Furthermore, the goal is to obtain the smallest possible number of statistically significant differences, as they are the least desired. We followed several methods to determine the sample size, the most important of which was the one described in a paper by Cade J et al. (2002) (Cade J, Thompson R, Burley V, Warm D., Development, validation and utilisation of food-frequency questionnaires – a review),

In addition, due to the use of the kappa coefficient, with a target power of the test of 80% for a minimum acceptable kappa of 0.40, an expected dropout rate of 10% and a significant p level of .05, the minimum sample size was n = 165. Conversely, with the inclusion of the calculated value of Cronbach’s ɑ for a minimum acceptable kappa of 0.70, a power of 80% and for a different number of items depending on the analysed nutritional index, the sample size ranged from 102 to 178. Therefore, we believe that the samples included in our analysis are enough to validate the questionnaire.

It should be noted that the study was conducted among young (the mean age of the participants is 21.0 and 21.6 in the "Slovakia" and "German" cohorts, respectively) and "healthy" adults (probably without chronic or profound metabolic disorders). In addition, the vast majority of the participants in the Slovakia cohort (approximately 90%) live relatively or very affluently. Therefore, the above might constitute a potential limitation of the study regarding the generalizability of its findings that should be acknowledged

Thank you for your very valuable remark. Indeed, significant, large differences were observed in the economic situation of the household between Slovakian and German students. We will be sure to use these findings in our next publication, which will present the topic of dietary habits and nutritional knowledge in the context of the assessment of household economic situation.

However, please bear in mind that the household economic situation was self-assessed. Consequently, the fact that the Slovakian students assessed their household situation higher (‘relatively affluent’ or ‘very affluent’) than the German students does not necessarily mean that their living conditions were actually better than their German peers’, as shown by some macroeconomic indicators obtained in Slovakia and Germany

Reviewer 2 Report

In this work the authors evaluate the reproducibility and reliability of the KomPAN questionnaire among groups of university students from Germany and Slovakia. Although, as they recognize, the validation of the questionnaires is essential, there are great deficiencies at the methodological level that prevent the understanding of the results. Fundamentally, the variables measured and the criteria used for the selection of the sample are not described.

These aspects and some others are detailed below:

ABSTRACT

1. The “Dietary Habits and Nutrition Beliefs Questionnaire (KomPAN)” is sometimes referred to as KomPAN and at other times as KOMPAN. It is recommended to unify.

2. “n=” can be omitted in the following sentence: “including n = 197 from Slovakia (men 26.2%) and n=225 from Germany (men 22.3%)…”

INTRODUCTION

3. In lines 37-41, the authors state that the use of questionnaires on food consumption makes it possible to assess food intake, but this is not only the case, since it also makes it possible to assess the intake of nutrients or food groups. So they can be a little more explicit.

4. It is recommended to end the introduction with the objective, so lines 57 to 59 related to the hypothesis and applicability of the results could be omitted.

MATERIAL AND METHODS

5. The inclusion of a flowchart summarizing the information related to the sample (lines 67-74) is recommended.

6. What were the student selection criteria?

7. Is this study part of another? If yes, what was the main objective of the study?

8. It is recommended to make a better description of the variables collected and their categorization in order to understand the results. For example, when is a subject considered to have a low level of “Physical activity during school or work”?

RESULTS

9. In relation to the Results Tables, it is recommended that the “ns” be replaced by the p-value, even if statistical significance is not reached. In addition, it is recommended to replace “sd” with “SD” and explain at the foot of the table all the abbreviations used.

10. It is striking that there are so many differences between the group of students from Germany and Slovakia (Table 1), what could this be due to? In addition, the authors do not describe anything about it, they only refer to it on line 115.

11. The authors refer to Table 2 twice, could they refer to it only once? On the one hand, lines 115-119 indicate the following: “The mean daily frequencies of consumption of most of the food products assessed in the questionnaire did not differ significantly between the test and the retest, both among the German students and the Slovakian students (Table 2).” However, in lines 121-126 the following appears: “In the German students, only the consumption of milk differed significantly between the test and the retest; while in the Slovakian students, only the consumption of white bread and fried foods differed significantly. The calculated mean values ​​for the Pro-Healthy-Diet-Index, Non-Healthy-Diet-Index and Total-Diet-Index and the intensity of the characteristics of the consumed products were similar between the test and the retest in both groups (Table two)."

DISCUSSION

12. I don't think it is necessary to refer to limitations as a separate section. The limitations and strengths of the study can be understood within the discussion.

Author Response

Dear Reviewer,

Thank you for reviewing our article. Below are our responses to each remark.

In this work the authors evaluate the reproducibility and reliability of the KomPAN questionnaire among groups of university students from Germany and Slovakia. Although, as they recognize, the validation of the questionnaires is essential, there are great deficiencies at the methodological level that prevent the understanding of the results. Fundamentally, the variables measured and the criteria used for the selection of the sample are not described. These aspects and some others are detailed below:

The “Dietary Habits and Nutrition Beliefs Questionnaire (KomPAN)” is sometimes referred to as KomPAN and at other times as KOMPAN. It is recommended to unify.

  • The article now uses ‘KomPAN’ consistently as the name of the questionnaire. Lines:15-16.

“n=” can be omitted in the following sentence: “including n = 197 from Slovakia (men 26.2%) and n=225 from Germany (men 22.3%)…”

        -      In lines 14–15 of the abstract, the ‘n’ has been removed. 

In lines 37-41, the authors state that the use of questionnaires on food consumption makes it possible to assess food intake, but this is not only the case, since it also makes it possible to assess the intake of nutrients or food groups. So they can be a little more explicit.

  • In lines 37-41 the following sentence has been added: Some questionnaires can also be used to assess the intake of nutrients or food groups [7-8]. Two references have been added. Additional papers have also been added to the References section.

It is recommended to end the introduction with the objective, so lines 57 to 59 related to the hypothesis and applicability of the results could be omitted. 

  • In lines 59-60, the hypothesis has been removed, as per the reviewer’s suggestion.

The inclusion of a flowchart summarizing the information related to the sample (lines 67-74) is recommended. 

  • As per the reviewer’s remark, a flowchart was created and added.

What were the student selection criteria? 

  • Thank you for your helpful remark. Indeed, we did not describe the inclusion criteria for the students. The criteria are now described in lines: 69-72. (Three inclusion criteria to the study were used: university students of medical courses except medicine and dietetics, declared willingness to participate in the test and retest and no disorders requiring a special diet.)

Is this study part of another? If yes, what was the main objective of the study?

  • The results presented in our study are part of a project involving the assessment of nutritional status and knowledge among university students attending courses related to public health. This information has been added to the manuscript. Part of the research is presented in the publication Suliga, E.; Cieśla, E.; Michel, S.; Kaducakova, H.; Martin, T.; Śliwiński, G.; Braun, A.; Izova, M.; Lehotska, M.; Kozieł, D.; Głuszek, S. Diet Quality Compared to the Nutritional Knowledge of Polish, German, and Slovakian University Students—Preliminary Research.  J. Environ. Res. Public Health2020, 17, 9062. https://doi.org/10.3390/ijerph17239062:  The aim of the study was to examine differences related to diet quality and level of knowledge among Polish, German and Slovakian university students, as well as to find out which factors differentiate their diet quality. Poland, Slovakia and the former GDR (German Democratic Republic) are neighbouring countries that underwent a similar systemic, economic and social transformation, which began in the 1980s. 

It is recommended to make a better description of the variables collected and their categorization in order to understand the results. For example, when is a subject considered to have a low level of “Physical activity during school or work”?

  • Thank you very much for your remark. The description of the research tool is now more detailed (lines:81-162), and we have changed the description of the categories in the tables. Below Table 1, we have added information about the categories of physical activity. Lines:206-214.

In relation to the Results Tables, it is recommended that the “ns” be replaced by the p-value, even if statistical significance is not reached. In addition, it is recommended to replace “sd” with “SD” and explain at the foot of the table all the abbreviations used.

  • We change sd into SD and explain all abbreviation below the tables.
  • We would like to keep the description of significance in the tables. There are a many information in our tables. Out tables contain a lot of numerical values. We think, that the additional numeric values will make tables less readable.
  • This way of presenting research result is accepted in scientific journals. (for example: owalkowska J, Wadolowska L, Czarnocinska J, Czlapka-Matyasik M, Galinski G, Jezewska-Zychowicz M, Bronkowska M, Dlugosz A, Loboda D, Wyka J. Reproducibility of a Questionnaire for Dietary Habits, Lifestyle and Nutrition Knowledge Assessment (KomPAN) in Polish Adolescents and Adults. Nutrients. 2018 Dec 1;10(12):1845. doi: 10.3390/nu10121845.)

It is striking that there are so many differences between the group of students from Germany and Slovakia (Table 1), what could this be due to? In addition, the authors do not describe anything about it, they only refer to it on line 115.

  • As per the reviewer’s suggestion, the description of Table 1 is now more detailed. Lines: 187-204. The difference between German and Slovak students depend on economic situation both countries and specificity of place of living. Slovakia there is rather an agricultural country with a small number of big cities. Rużomberok (about 28000 inhabitants) is situated in the mountains and surrounded a large number of small villages. while Brandenburg (include Cottbus and Senftenberg towns) is a region of Germany with developed industry and agriculture. Despite the economic differences, Slovak students have a higher sense of quality of life. May be due to the fact that the majority of students in Slovakia live with their parents, while German students live in dormitories, and more often they support themselves.

I don't think it is necessary to refer to limitations as a separate section. The limitations and strengths of the study can be understood within the discussion.

  • Limitations have been included in the discussion. Line:388.

Round 2

Reviewer 1 Report

Manuscript ID: nutrients- 2012028 (Revised version)

Authors: Elżbieta Cieśla et al.

Title: "Reproducibility of the German and Slovakian versions of the Dietary Habits and Nutrition Beliefs Questionnaire (KomPAN)"

The authors of the present study tried to address my comments and revise the manuscript. However, for a few comments, the authors provided an answer in their reply, but they did not update the manuscript accordingly.

The following comments should be considered:

1.     The text and the numbers are unreadable in some boxes of the flowchart, at least in my PDF.

2.     Since the authors have performed a power and sample size calculation, it would further improve the paper and be helpful for future studies to add a relevant comment in the statistical analysis.

3.     As far as my following comment is concerned: "It should be noted that the study was conducted among young (the mean age of the participants is 21.0 and 21.6 in the "Slovakia" and "German" cohorts, respectively) and "healthy" adults (probably without chronic or profound metabolic disorders). In addition, the vast majority of the participants in the Slovakia cohort (approximately 90%) live relatively or very affluently. Therefore, the above might constitute a potential limitation of the study regarding the generalizability of its findings that should be acknowledged.". Since the authors agree with the above comment regarding the characteristics of the study population (therefore generalizability issues) and the differences between the two examined groups, the authors should discuss it and include it as a potential limitation of the current study, as mentioned/described above.

4.     I would suggest replacing the mode with the median. Preferably, since the number of persons per family ranges from 2 to 5, the authors could present the absolute numbers for each subcategory (i.e., from 2 to 5) in separate rows in Table 1.  

5.     I would suggest replacing the text "(quartile 1 –quartile 3)" with the interquartile range (IQR).

Author Response

Dear Reviewer,

Thank you for taking the time to review our article and for your important feedback, which will help to improve the scientific value of the article.  Below are our responses to each remark.

  1.  The text and the numbers are unreadable in some boxes of the flowchart, at least in my PDF: we corrected flowchart 1. 
  2. Since the authors have performed a power and sample size calculation, it would further improve the paper and be helpful for future studies to add a relevant comment in the statistical analysis:  we added few sentences about sample size calculation: lines: 205-208 
  3. As far as my following comment is concerned: "It should be noted that the study was conducted among young (the mean age of the participants is 21.0 and 21.6 in the "Slovakia" and "German" cohorts, respectively) and "healthy" adults (probably without chronic or profound metabolic disorders). In addition, the vast majority of the participants in the Slovakia cohort (approximately 90%) live relatively or very affluently. Therefore, the above might constitute a potential limitation of the study regarding the generalizability of its findings that should be acknowledged.". Since the authors agree with the above comment regarding the characteristics of the study population (therefore generalizability issues) and the differences between the two examined groups, the authors should discuss it and include it as a potential limitation of the current study, as mentioned/described above: We added explanation - lines: 424-429.
  4.  I would suggest replacing the mode with the median. Preferably, since the number of persons per family ranges from 2 to 5, the authors could present the absolute numbers for each subcategory (i.e., from 2 to 5) in separate rows in Table 1 - we replaced Mo(min-max) with the median(IQR)-Table 1: Number person in family
  5. I would suggest replacing the text "(quartile 1 –quartile 3)" with the interquartile range (IQR): - we relaced (quartile 1-quartile 3) with (IQR).

Reviewer 2 Report

The manuscript has improved considerably. However, it would be necessary to address the following point before proceeding with the publication:

- the flowchart must appear as "Figure 1" throughout the manuscript and at the bottom of it. Along with it, this cannot be read in its entirety. Some of the frames are cut, not being possible to continue the same. Can the authors correct it?

Author Response

Dear Reviewer,

Thank you for taking the time to review our article and for your important feedback, which will help to improve the scientific value of the article.  Below are our responses to each remark.

1.  the flowchart must appear as "Figure 1" throughout the manuscript and at the bottom of it. Along with it, this cannot be read in its entirety. Some of the frames are cut, not being possible to continue the same. Can the authors correct it?  the flowchart was corrected. 
